# Recent Trends in Active Packaging Using Nanotechnology to Inhibit Oxidation and Microbiological Growth in Muscle Foods

**DOI:** 10.3390/foods12193662

**Published:** 2023-10-04

**Authors:** Rickyn A. Jacinto-Valderrama, Cristina T. Andrade, Mirian Pateiro, José M. Lorenzo, Carlos Adam Conte-Junior

**Affiliations:** 1Programa de Pós-Graduação em Ciência de Alimentos, Instituto de Química, Universidade Federal do Rio de Janeiro, Centro de Tecnologia, Avenida Athos da Silveira Ramos 149, Rio de Janeiro 21941-909, RJ, Brazil; rickynjacinto@gmail.com (R.A.J.-V.); ctristaoandrade@gmail.com (C.T.A.); 2Centro Tecnológico de la Carne de Galicia, Avd. Galicia n° 4, Parque Tecnológico de Galicia, San Cibrao das Viñas, 32900 Ourense, Spain; mirianpateiro@ceteca.net (M.P.); jmlorenzo@ceteca.net (J.M.L.); 3Área de Tecnología de los Alimentos, Facultad de Ciencias de Ourense, Universidad de Vigo, 32004 Ourense, Spain

**Keywords:** food spoilage, meat, active packaging, natural antimicrobials, natural antioxidant, nanocarriers

## Abstract

Muscle foods are highly perishable products that require the use of additives to inhibit lipid and protein oxidation and/or the growth of spoilage and pathogenic microorganisms. The reduction or replacement of additives used in the food industry is a current trend that requires the support of active-packaging technology to overcome novel challenges in muscle-food preservation. Several nano-sized active substances incorporated in the polymeric matrix of muscle-food packaging were discussed (nanocarriers and nanoparticles of essential oils, metal oxide, extracts, enzymes, bioactive peptides, surfactants, and bacteriophages). In addition, the extension of the shelf life and the inhibitory effects of oxidation and microbial growth obtained during storage were also extensively revised. The use of active packaging in muscle foods to inhibit oxidation and microbial growth is an alternative in the development of clean-label meat and meat products. Although the studies presented serve as a basis for future research, it is important to emphasize the importance of carrying out detailed studies of the possible migration of potentially toxic additives, incorporated in active packaging developed for muscle foods under different storage conditions.

## 1. Introduction

Muscle foods are an important source of essential nutrients for diets, such as proteins of high biological value, lipids, complex B vitamins, and essential minerals [1,2]. Oxidation and microbial growth are the main causes of quality loss during the storage of muscle foods, influencing the sensory characteristics of the products [3,4].

Lipid oxidation occurs when unsaturated fatty acids react with pro-oxidant components, e.g., molecular oxygen (O_2_), metal ions (Fe^2+^, Cu^+^), light, and pro-oxidant enzymes, which induce the onset of the oxidative reaction [5,6]. In this regard, a reduction in the nutritional value of muscle foods may be due to the loss of essential fatty acids, as well as the loss of sensory quality. This reaction generates volatile compounds related to off-flavors (rancidity), and the production of various toxic compounds, like aldehyde derivatives. Such substances increase the risk of developing degenerative and chronic diseases (e.g., atherosclerosis, cancer, inflammation, and aging processes, among others) [7,8]. The most common way to measure the degree of lipid oxidation in muscle foods is through the peroxide value (POV) and thiobarbituric acid reactive substances (TBARs) tests, which measure the content of hydroperoxides and malondialdehyde, respectively [9].

Protein oxidation occurs when muscle proteins react with reactive oxygen species; for instance, superoxide, hydroperoxyl, and other non-radical species like hydroperoxides, recognized as potential initiators [3,10]. Physical factors (e.g., light and irradiation), and other chemical species (radicals, Fe^2+^ and Cu^+^ metal ions, reducing sugars, and pro-oxidative heme proteins like myoglobin) may also cause oxidation reactions [11]. Protein oxidation can lead to changes in the physical properties of proteins, including fragmentation, aggregation, loss of solubility, functionality, and decreased susceptibility to proteolysis. All these changes are related to the development of undesirable flavors and discoloration of muscle foods [4,12]. These sensorial changes make meat products unacceptable to consumers. Oxidative degradation and the production of carbonyl compounds, cross-linking (disulfide bonds formation), or the production of sulfur-containing derivatives (sulfone, sulfoxide, and disulfide derivatives) modifies protein digestibility, which negatively affects the nutritional value of meat products [13,14]. Protein oxidation can be determined as the concentration of the thiol group in muscle foods through Ellman’s reagent (5,5′-dithiobis-(2-nitrobenzoic acid) or DTNB), mass spectrometry, carbonyl, and metmyoglobin content [11].

Meat and meat products are excellent culture medium for a variety of bacteria, yeasts, and molds, some of them pathogens [15,16]. They are responsible for quality loss, triggering changes in color, texture, flavor, and oxidation in stored muscle foods. Thereby, the biological activity of some microorganisms acts as a pro-oxidant using O_2_ to begin the oxidation of myoglobin to metmyoglobin. This reaction is related to the undesirable discoloration in muscle foods [17]. In contrast, some bacteria have shown the ability to improve meat color by converting metmyoglobin to oxymyoglobin [18]. Meat protein degradation, caused by microbial and enzymatic activities, is broken down into methylamines and ammonia [19]. Because of this reaction, the quantification of total volatile basic nitrogen (TVB-N) is commonly used to determine the freshness of muscle foods.

In the last decade, the consumption of meat and meat products has been associated with diseases like obesity, hypertension, coronary heart disease, and microbiological risks [20,21]. In addition, the International Cancer Research Agency (IARC), which is part of the United Nations World Health Organization, has classified meat products as carcinogenic to humans [22]. In this sense, consumers demand “healthier” or “clean label” foods, which present a reduction or replacement of saturated fats, sodium, nitrites, phosphates, and other additives [23,24]. However, the reduction or replacement of additives can affect the shelf life of muscle foods, initiating lipid oxidation, protein oxidation, and microbial growth [16,17,18,19,20,21,22,23,24,25].

In contrast, the amount of antioxidants and antimicrobials used in food processing may be even greater, due to the size of the area of contact with the food matrix [26]. However, in most cases, the degradation and decomposition of muscle foods start on the surface, where the exposure to molecular oxygen is greater. It is also on the surface that the development of aerobic microbial spoilage by *Pseudomonas*, *Acinetobacter*, *Moraxella*, *Acetobacter*, etc., begins [4]. Therefore, depositing active substances to the surface of meat products or adding these substances to the packaging film reduces the quantity of additives necessary to avoid oxidation or microbial spoilage [27,28].

In recent decades, active packaging has been developed to purposefully interact with food to prolong its shelf life, ensuring quality, safety, and integrity [28,29,30,31]. Two types of active packaging modalities were developed: (i) active scavenging systems, which induce a desirable response from food systems without the active component migrating from packaging to food, and (ii) active releasing systems, which may allow a controlled migration of non-volatile compounds or an emission of volatile compounds into the atmosphere around the food [26].

To extend the shelf life of packaged muscle foods, pads or sachets are used inside the package to absorb moisture, and release carbon dioxide and other active compounds [32,33]. Nonetheless, this modality has been little accepted by the consumer [34]. Another alternative is the direct incorporation of active or chemoactive substances in the polymeric matrix through melt extrusion (film), or on the surface of the film (coating) [35,36]. The nature of these substances is varied. For example, essential oils (EO), oleoresins (OR), plant extracts (PE), bacteriostatics, phenolics, polysaccharides, surfactants, bacteriophages, metals, and metal oxides have been incorporated to obtain active packaging (Figure 1).

Recently, consumers have shown an increasing interest in the substitution of synthetic additives, widely used by the food industry, for natural additives with antioxidant and/or antimicrobial activity. This interest has demanded extensive research on natural additives with no side effects on consumer health [31,37,38]. However, the quantity used will influence the active power, desired effect time, and sensory quality, being able to modify the characteristic flavor in muscle foods, if applied at inappropriate levels [35]. One of the advantages of active packaging is the possibility of tuning the amount of active additive that reaches the food surface [33].

Another issue to be considered regarding food packaging consists of the nature of the polymer film. To achieve environmental requirements, the use of natural polysaccharides and proteins as matrices of food packaging is recommended [27,39]. However, the mechanical and barrier properties of biopolymers present disadvantages, mainly low tensile strength, and high hydrophilicity. Such properties hinder their use in packaging for muscle foods [40,41]. Nevertheless, the addition of fillers, with high resistance and functional properties, has attracted the interest of researchers. Graphene nanoparticles may be cited as an example of filler [42,43].

Nanotechnology can be defined as the study, fabrication, and application of structures with nanoscale dimensions (1 nm–100 nm) in at least one of the dimensions [44]. Nano-scale materials have specific properties. When added to food packaging, some properties are significantly improved and their application can bring advantages [45,46]. Its application in active packaging can bring great advantages due to the larger surface area of nanomaterials, allowing greater interaction with muscle foods [47,48]. Nanoparticles (NPs) are prepared by a variety of chemical and physical methods that are relatively expensive, and potentially hazardous to the environment [44]. However, bioactive NPs are considered promising materials [29]. In this sense, much research has been drawing the attention of the food industry to develop safe, active antimicrobial packaging that contributes to the reduction in additives in the preservation of muscle foods [49].

Furthermore, the incorporation of nanocarriers (nanophytosomes, nanoliposomes, and nanoemulsions) in the polymer matrix to improve antioxidant and antimicrobial properties has been the subject of recent studies (Figure 2) [50]. Nanophytosomes (NPhs) are nanoparticles consisting of encapsulated phytoactive compounds in phospholipids. NPhs generate water and lipid-soluble complexes [51]. Nanoliposomes (NLs) are colloidal vesicles of nanometric size (<200 nm) constituted of phospholipid bilayers, which surround the bioactive aqueous or hydrophobic material. Nanoemulsions (NEs) are colloidal dispersions comprising two immiscible liquids with droplet sizes ranging from 50 to 1000 nm, in which one is dispersed in the other [52,53].

This work is an overview of studies published in recent years on active packaging with nanoparticles and nanocarriers of antioxidant and antimicrobial substances applied to muscle-foods. This review covered studies that show the inhibitory effect on the oxidation of lipids and proteins and on the development of spoilage and pathogenic microorganisms. Although the packaging of antioxidant active ingredients has been widely studied, as is the case with essential oils and plant extracts, this work aimed to show how nanotechnology can help preserve the active ingredients applied and the effect on the shelf life of muscle foods. Furthermore, a subject that has been little studied to date was addressed, such as the application of Maillard reaction products, surfactants, and bacteriophages, among other antimicrobials applied in active packaging for muscle foods. Therefore, the aim of this review article was to understand how unconventional active packaging is innovating to slow or inhibit oxidation and microbial growth in muscle foods.

## 2. Active Packaging in Muscle Foods

### 2.1. Active Packaging with Essential Oils, Oleoresins, and Their Nanocarriers

EO are secondary metabolites that defend the plant from attacks by insects and microorganisms. In addition, most of these EO are classified as generally recognized as safe (GRAS), which is why EO have been widely studied as additives in active films and coatings [54]. OR are viscous liquids extracted from spices, which contain not only essential oil but also some non-volatile components, such as pigment, fatty oils, and phenolic antioxidants [35].

There are several scientific studies that prove the antioxidant and antimicrobial activity of EO and OR incorporated in active packaging. Nevertheless, this technology has been little used by the food industry. Aldehydes, phenols, and oxygenated terpenoids are the main hydrophobic components responsible for the antimicrobial activity of EOs and ORs (Figure 3) [55,56]. These components interact with the microbial cell membrane, due to the loss of essential cellular molecules, such as polysaccharides, fatty acids, and phospholipids, making the structures more permeable [57]. In addition, once within the cell, they interfere with the normal activity of the cell membrane (electron transfer, nutrient exchange, protein synthesis, nucleic acids, and enzymatic activity) and interact with cellular organelles (ribosomes, endoplasmic reticulum, and Golgi body), affecting the respiratory processes leading to cell death (Figure 4) [58,59]. On the other hand, EOs and ORs have numerous modes of action (direct or indirect) to inhibit the oxidative reactions preventing the formation of free radicals, chelating the metal ions, and interrupting the propagation of free radicals through scavenging species produced in oxidative reactions [60,61].

The direct incorporation of EO in the polymeric matrix is the most common way to obtain active packaging with antioxidant and antimicrobial properties. Table 1 shows some studies where EO was applied in active packaging for muscle foods.

Non-biodegradable active packaging embedded with OR has been used to inhibit the oxidation, and microbial growth of foods. In this sense, Song et al. [35] incorporated 8% rosemary oleoresin to coat polyethylene terephthalate (PET) film in order to extend the shelf life of ground pork stored at 4 °C for 14 days. The results indicated that the addition of 8% rosemary OR inhibited lipid oxidation (TBARs) and protein degradation (TVB-N) of samples compared to control with treatments with 6% rosemary oleoresin. Alternatively, Laorenza and Harnkarnsujarit [36] studied the inhibitory effect of biodegradable active packaging (PBAT/PLA) incorporated with ginger and lemon peel essential oil obtained by extrusion. The results showed that there was inhibition against *Bacillus cereus* and a reduction in the total viable count.

In the research by Zhang et al. [62], the active biodegradable active film was developed to prolong the shelf life of pork meat during storage at 4 °C. The active film was produced by mixing poly(vinyl alcohol) (PVA) and curdlan gum, and 0%, 1%, 1.5%, and 2% thyme EO. The results showed that the higher concentration of thyme EO in the film decreased lipid oxidation, protein degradation, and microbiological growth when compared to the control treatment. The incorporation of thyme EO in active film extended the shelf life of pork meat from 12 to 16 days (TBARs), from 10 to 16 days (TVB-N), and from 8 to 14 days (total viable count (TVC), and *Escherichia coli*).

On the other hand, Bharti et al. [63] examined the effect of *Manihot esculenta* starch and carrageenan film functionalized with 0.5% of anise EO, 1% of caraway EO, or 1% of nutmeg EO in aerobically wrapped chicken nuggets storage at 4 °C for 15 days. The results indicated that lipid oxidation (POV and TBARs) in treated samples was significantly (*p* < 0.05) lower than in control samples. Additionally, the TVC, psychrophilic count (PPC), yeast, and mold count (YMC) were also significantly (*p* < 0.01) lower in treatment groups and were within the permissible limits, extending the shelf life from 12 to 15 days.

Another important study including polysaccharides in active packaging was the research by Cao and Song [64], who developed *Bombacaceae* gum with the incorporation of 1.25% of cinnamon leaf EO to preserve the shelf life of fresh salmon fillet storage at 4 °C for 15 days. *Bombacaceae* gum contains phenolic compounds, especially phenolic acids, catechol tannin, gallic acid, and tannic acid, that give its functional properties. This study indicated that the incorporation of 1.25% of cinnamon leaf EO in *Bombacaceae* gum film empowered the inhibition of lipid oxidation in fresh salmon fillets during storage, prolonging the shelf life from 3 to 15 days (POV) and from 6 to 15 days (TBARs).

On the other hand, it is possible to use natural polymers with inherent antioxidant and antimicrobial activity to develop active food films. Chitosan is a polysaccharide obtained by the deacetylation of chitin, which is the second most abundant polysaccharide found in nature after cellulose. The chitosan antimicrobial activity is due to the presence of positively charged amino groups that interact with negatively charged macromolecules on the microbial cell surface, disrupting the lipopolysaccharide layer of the outer membrane of Gram-negative bacteria, and being a barrier against oxygen transfer [65]. According to Ehsani et al. [66], the shelf life of fish burgers wrapped in chitosan film functionalized with 0.5% sage EO was extended during storage at 4 °C for 20 days. Deterioration by TVC, psychrotrophic bacteria count (PTC), *Pseudomonas* spp. count (PC), *Shewanella* spp. count, and oxidative rancidity (TBARs) were inhibited during the storage.

The antimicrobial properties in active packaging embedded with EO can be reinforced with montmorillonite (MMT) [67]. In this regard, Pires et al. [68] demonstrated that the incorporation of 2% of rosemary or ginger EO in active film of chitosan reinforced with 2.5% MMT inhibited the lipid oxidation of chicken meat stored for 15 days at 5 °C compared to chitosan film and chitosan film reinforced with MMT. Moreover, it does not improve the antimicrobial activity against TVC and total coliform bacteria (TCB).

Edible coatings are applied directly onto the food surface from liquid suspension, emulsion, or powder form [69]. Additionally, an adhesion process requires diffusion between both the coating solution and the surface area of the food product [70]. Thereby, Mehdizadeh, and Langroodi [71] developed an edible coating package of chitosan with the incorporation of 1% of *Zataria multiflora* EO to prolong the shelf life of chicken breast meat stored for 16 days at 4 °C. The results indicated that the chitosan-active coating extended the shelf life from 8 to 16 days for lipid oxidation (TBARs) and protein degradation (TVB-N). The shelf life of microbial quality was extended from 4 to 16 days for TVC and lactic acid bacteria count (LABC), and from 4 to 8 days for PC.

**Table 1 foods-12-03662-t001:** Application of essential oil, oleoresin, metal, metal oxide, and their nanocarriers in active packaging for muscle foods.

Meat/Meat Products	Essential Oil and Metal-Oxide	Polymeric Matrix	Inhibitory Effect Against	Reference
Lamb meat	TiO_2_ NPs-rosemary essential oil	Whey protein isolate film	Lipid oxidation, protein degradation, and PTC	[27]
Minced pork meat	Rosemary oleoresin	Coating of PET film	Lipid oxidation and protein degradation	[35]
Shrimp	Ginger essential oil	PBAT/PLA film	TVC, *Bacillus cereus*	[36]
Chicken sausage	Ag NPs	PVA/MMT film	TVC	[45]
Ready-to-eat Yao pork meat	Star anise essential oil and antimicrobial mixture NEs	*Artemisia sphaerocephala* Krasch. gum coating	Protein degradation, TVC and *Escherichia coli*	[47]
Chicken meat	ZnO NPs	Carboxymethyl cellulose film	Lipid oxidation, protein degradation, TVC, LABC, and *Staphylococcus aureus*	[48]
Pork meat	Thyme essential oil	Curdlan and PVA film	Lipid oxidation, protein degradation, and TVC, and *Escherichia coli*	[62]
Chicken nugget	Nutmeg essential oil	*Manihot esculenta* starch/carrageenan film	Lipid oxidation, TVC, PPC, and YMC	[63]
Salmon fillets	Cinnamon leaf essential oil	Bombacaceae gum film	Primary oxidative products	[64]
Fish burger	Sage essential oil	Chitosan film	TVC, PTC, *Pseudomonas* spp., and *Shewanella* spp	[66]
Poultry meat	Rosemary essential oil	Chitosan/MMT film	Lipid oxidation, TVC, and TCB	[68]
Poultry meat	Ginger essential oil	Chitosan/MMT film	Lipid oxidation, TVC, and TCB	[68]
Chicken meat	*Zataria multiflora* essential oil	Chitosan coating	Lipid oxidation, protein degradation, TVC, PPC, LABC, and *Pseudomonas* spp.	[71]
Chicken meat	Black cumin essential oil	Chitosan/alginate multilayer film	TVC and PTC	[72]
Shrimp	Cinnamon essential oil NPhs	PVA/boric acid film	TBC, *Pseudomonas aeruginosa*	[73]
Lamb meat	*Satureja* plant essential oil NLs	Chitosan coating	Lipid oxidation, TVC, LABC, and *Pseudomonas* spp.	[74]
Cooked sausage	Garlic essential oil NLs	Chitosan or whey protein film	Lipid oxidation, APC, PTC, and LABC	[75]
Beef	TiO_2_ NTs	Whey protein nanofibrils film	Lipid oxidation and TVC	[76]
Pork meat	ZnO NPs	Chitosan film	TBC	[77]
Pork meat	Ag NPs-laurel essential oil NLs	PE/chitosan coating	Protein degradation	[78]
Smoked salmon	ZnO NPs	Gracilaria vermiculophylla agar film	*Salmonella* Typhimurium	[79]

APC, aerobic plate count; LABC, lactic acid bacteria count; MMT, montmorillonite; NEs, nanoemulsions; NLs, nanoliposomes; NPs, nanoparticles; NPhs, nanophytosome; NTs, nanotubes; PE, polyethylene; PET, polyethylene terephthalate; PPC, psychrophilic count; PTC, psychrotrophic count; PVA, polyvinyl alcohol; TBC, total bacterial count; TCB, total coliform bacteria; TVC, total viable count; YMC, yeast and mold count.

Multilayered packaging or composite materials are a preferred packaging option due to many advantages, such as higher toughness and tensile strength compared to the individual ingredients [70]. Multilayered polymer films bound to layers that incorporate active compounds have gained attention in the food-packaging industry. In this sense, Takma and Korel [72] developed a multilayer PET film with 10 layers of chitosan and alginate embedded with black cumin EO to prolong the shelf life of chicken meat during the storage for 5 days at 4 °C. The outer layer of coated PET film was an alginate solution with 1% of black cumin EO that was in contact with samples. There was antimicrobial activity against TVC and PTC, extending the shelf life of chicken meat.

Essential oils are sensitive to light, oxygen, and temperature, and are highly volatile when applied in active packaging [33]. In contrast, nanocarriers (NPhs, NLs, and NEs) of essential oils have been developed to protect these bioactive compounds from degradation. Nazari et al. [73] conducted a study to evaluate the effects of incorporating 5% of cinnamon EO NPhs into cross-linked polyvinyl alcohol (PVA) nanofiber and boric acid films using electrospinning technique to improve the antimicrobial activity in shrimp storage for 7 days at 4 °C. The results indicated that TBC and *Pseudomonas aeruginosa* count (PAC) were reduced in samples wrapped in active films compared to the control and with the incorporation of cinnamon EO treatments.

Moreover, Pabast et al. [74] inhibited microbial growth and lipid oxidation in lamb meat with chitosan coating functionalized with 1% of *Satureja* plant EO NLs during the storage for 20 days at 4 °C. Microbial spoilage by TVC, PC, LABC, and lipid oxidation was inhibited by 10 days in meat samples coated with the active solution. Similar effects were demonstrated by Esmaeili et al. [75], who observed an inhibition of microbial spoilage (TVC, PTC, and LABC) and lipid oxidation (POV and TBARs) in cooked sausage wrapped in chitosan or whey protein film embedded with 2% of garlic EO NLs during the storage for 50 days at 4 °C.

Moreover, a synergistic effect was observed by mixing EO with other antioxidants and/or antimicrobials, especially using NEs. In this regard, Liu et al. [47] successfully prolonged the shelf life of Yao ready-to-eat pork meat coated with *Artemisia sphaerocephala* Krasch. gum film functionalized with 0.6% NEs (star anise EO, polylysine, and nisin). The result demonstrated that the shelf life was extended from 8 to 20 days by TVC, and from 12 to 20 days by *Escherichia coli*. Additionally, the protein degradation (TVB-N) was inhibited during the storage compared to the control containing only star anise EO and negative control.

### 2.2. Active Packaging with Metal and Metal Oxide Nanoparticles

The antibacterial property of metal and metal oxide NPs depends directly on the particle size and shape. In addition, these NPS can affect the cell membrane permeability and interact with cellular organelles, resulting in genotoxicity and cell death [80]. The development of green experimental processes for the synthesis of metal NPs has been of importance because it is non-toxic, cost-effective, and environmentally friendly. Nowadays, green metal NPs synthesis can use green plants that contain several biomolecules, e.g., alkaloids, flavonoids, and terpenes, which act as reducing as well as capping agents [45]. Therefore, it is the safest method for the development of active food packaging. Table 1 shows some active packaging for muscle foods functionalized with metal NPs.

Silver nanoparticles (Ag NPs) are one of the most used nanoparticles in the industry due to their excellent antimicrobial potential against multidrug-resistant pathogens, low toxicity to human cells, and thermally stable [81]. When the Ag NPs incorporated in the polymeric matrix undergo oxidative processes in an aqueous environment, silver ions (Ag^+^) are released. These ions can interact with atoms that have a higher electron density, encouraging the permeability and interruption of bacterial metabolism [82]. Mathew et al. [45] successfully developed biodegradable and active PVA-MMT films with *in situ* generated ginger extract mediated Ag NPs to reduce microbial growth during the refrigerated storage of chicken sausages. A photo-assisted method using sunlight irradiation was adopted for the rapid and eco-friendly in situ generation of ginger extract mediated Ag NPs in the composite from AgNO_3_. After 4 days of incubation of samples at 4 °C, the chicken sausages wrapped in PVA-MMT-Ag NPs film indicated lower growth of TVC compared to control samples.

Zinc oxide is another additive used in the food industry and is classified as GRAS by FDA due to non-toxicity and high stability [54]. In active food packaging, zinc oxide nanoparticles (ZnO NPs) are widely known as antimicrobials. When reactive oxygen species (ROS) are exposed to ZnO NPs, it causes the release of Zn^2+^ ions into the aqueous environment. When Zn^2+^ ions are at high concentrations, they interact with the negatively charged cell membrane, leading to leakage to death through a cytotoxic mechanism [83]. Therefore, Wang et al. [77] developed a chitosan-based film functionalized with ZnO NPs to wrap pork meat during storage at 4 °C for 7 days. The result showed that there was a reduction in the speed of growth of TBC compared to the control treatment. On the other hand, Mohammadi et al. [48] demonstrated that the addition of 1% of ZnO NPs in active packaging is not only limited to the reduction in microbial growth of TVC, LABC, and *Staphylococcus aureus* during storage. In addition, it inhibits lipid oxidation and protein degradation of chicken meat samples wrapped in carboxymethyl cellulose film stored at 4 °C for 9 days. In addition, Baek and Song [79] observed a similar result when incorporating 3% of ZnO NPs in *Gracilaria vermiculophylla* agar-based film to prolong the shelf life of smoked salmon stored for 9 days at 4 °C. This novel active packaging inhibited the microbial growth of *Listeria monocytogenes* and *Salmonella* Typhimurium extending the shelf life. POV and TBARs values were the lowest among the treatments during the storage for 9 days.

Titanium dioxide was authorized for use as a food additive in the European Union [84]. The antimicrobial activity of titanium dioxide nanoparticles (TiO_2_ NPs) can be related to several mechanisms: interaction directly with the microbial cells, production of oxidative secondary products, or dissolution of heavy metal ions that cause damage [85]. The application of TiO_2_ NPs and titanium dioxide nanotubes (TiO_2_ NTs) in active food packaging has been characterized by their low toxicity to human cells, potential as a UV absorber, high thermal stability, and antimicrobial activity [86]. In this sense, Feng et al. [76] compared the antioxidant and antimicrobial activity of the addition of TiO_2_ NPs and TiO_2_ NTs in whey protein nanofibrils-based edible coating film to extend the shelf life of beef stored at 4 °C for 15 days. There was no significant difference in the TBARs values during the storage between beef samples coating film functionalized with TiO_2_ NPs and TiO_2_ NTs, but there was a difference compared to the control sample. Nonetheless, the TVC of samples coating film with TiO_2_ NTs presented the slowest counts after 15 days compared to control and coating film with TiO_2_ NPs sample.

On the other hand, some studies showed that multi-component active films can increase the quality and shelf life of muscle foods (Figure 5). In this regard, Alizadeh-Sani et al. [27] successfully developed a biodegradable active packaging from whey protein isolate film embedded with cellulose nanofiber, as a filler, 1% of TiO_2_ NPs and 2% of rosemary oil EO, as antimicrobial and antioxidant agents, to preserve lamb meat stored for 15 days at 4 °C. The result indicated that microbial growth of PTC and lipid oxidation (POV and TBARs) in lamb meat were significantly reduced during storage, extending the shelf life from around 6 to 15 days. Similar effects were demonstrated by Wu et al. [78] developing a PE film coated with chitosan functionalized with NLs encapsulating laurel EO and Ag NPs to extend the shelf life of pork meat during storage at 4 °C for 15 days. The result showed that control samples wrapped in PE film prolonged the shelf life over 9 days; meanwhile, packaged samples with multi-component active films (laurel EO and Ag NPs nanoliposomes) became inedible after 15 days.

### 2.3. Active Packaging with Plant Extracts and Their Nanocarriers

Plant extracts (PEs) are substances characterized by their antioxidant and antimicrobial activity, including glycosides, polyphenols, alkaloids, and others [87,88]. These compounds can increase the permeability of the cell membrane of microorganisms, and interfere with cell metabolism, leading to cell death [58]. Moreover, PEs have the property of scavenging free radicals produced in oxidative reactions during the storage of muscle foods [89].

The incorporation of plant extracts in films to produce functional active packaging and preserve the qualities of muscle foods has recently attracted increased interest [26]. Table 2 shows plant extract-based active packaging for muscle foods.

Betalains are plant pigments with antioxidant activity widely used in the industry due to their high pH stability, generating different colors ranging from red to yellow, for being non-toxic, and reducing the risk of developing degenerative diseases [104,105]. Kanatt [88] examined the effect of incorporation of *Amaranthus* leaf extract (rich in betalains) in PVA–gelatin film to preserve the microbiological quality (TBC, *Staphylococcus aureus,* and fecal coliforms), oxidative rancidity (TBARs), and protein degradation (TVB-N) of minced chicken, and fish fillets during the storage at 2 to 4 °C for 12 days. The result indicated that control samples had a shelf life of 3 days, whereas samples wrapped in active films deteriorated only after 12 days.

Phenolic compounds constitute another large group of active compounds widely used in the development of active food packaging. Phenolic compounds, also called polyphenols, are secondary metabolites of plants with antioxidant and antimicrobial activity [89,106]. Hydroxyl groups of these substances are known to donate hydrogen and intercept the free radical oxidation chain, thereby reducing lipid oxidation [107]. Maru et al. [90] extended the shelf life of chicken meat using pullulan or chitosan coating, both with 1% lemon peel polyphenol extract. The microbiological growth of TBC and *Enterobacteriaceae* and lipid oxidation were inhibited. The shelf life of chicken meat was extended to 6 days using pullulan coating, and to 14 days using chitosan coating compared to control with 1 day. Similar results were obtained by Borzi et al. [94], who developed a polyamide (nylon 6)-active film with the addition of 5% of green tea extract (rich in polyphenols, mainly in catechins), to extend the shelf life of minced beef during the storage for 23 days at 4 °C. The results showed that lipid (TBARs) and protein oxidation (metmyoglobin content), and protein degradation (TVB-N) in samples wrapped with the active film were inhibited during the storage, prolonging the shelf life of minced beef.

On the other hand, some studies showed that a synergistic effect occurred when mixing plant polyphenols extract with other natural antioxidants and antimicrobials. Sogut and Seydim [91] developed a chitosan film embedded with 15% grape seed extract (rich in polyphenols, mainly in proanthocyanidins) and 2% of nanocellulose to prolong the shelf life of chicken fillets during storage for 15 days at 4 °C. The result demonstrated that the shelf life of chicken fillets was extended from 9 to 12 days by TBARs, and from 6 to 15 days by TVC, and TCB for acceptable values compared to negative and positive control. A similar result was demonstrated by Chollakup et al. [95] extending the microbiological quality of sliced salami wrapped in whey protein isolate film from 5 to 10 days during storage at room temperature. This active film was functionalized with 0.02% rambutan peel extract (rich in polyphenols) and 0.02% cinnamon EO. The shelf life of sliced salami was prolonged from 5 to 10 days by TVC. In contrast, Xiong et al. [93] found no synergistic effect in the incorporation of 0.5% grape seed extract and 0.1% of nisin in chitosan–gelatin edible coating of pork meat stored for 20 days at 4 °C. The result showed that chitosan–gelatin coating functionalized with grape seed extract had the best performance on pork preservation (TBARs value and DTNB test) compared to control and active coating with incorporation of grape seed extract and nisin during the storage.

Polyphenols are sensitive to light, oxygen, and temperature when applied in active packaging [107]. Moreover, nanocarriers such as nanoencapsulation can inhibit the oxidation of plant extract. In this sense, Cui et al. [92] applied chitosan nanoparticles encapsulated with an extract of pomegranate peel (rich in polyphenols) in zein film to prolong the shelf life of pork stored for 12 days at 4 °C. The results revealed that in samples wrapped in active zein film, there was a reduction in *Listeria monocytogenes* during storage.

### 2.4. Active Packaging with Other Active Compounds and Their Nanocarriers

Active compounds are widely used in the meat industry to improve the nutritional, physicochemical properties and prolong the shelf life of muscle foods [31]. Several active compounds can be incorporated including synthetic and natural compounds (e.g., sodium chloride, vitamins, phosphates, antioxidants, and antimicrobials). The incorporation of active compounds in a polymeric matrix to extend the shelf life of muscle foods is an alternative used in the meat industry [26]. Table 2 shows active packaging functionalized with active compounds in muscle foods.

Sodium nitrite (NaNO_2_) is an additive commonly used to prolong the redness in meat products. However, high levels of nitrite application can cause adverse safety hazards [108]. The recent research has investigated alternative processes to replace or reduce the direct addition of nitrite compounds into meat products. Chatkitanan and Harnkarnsujarit [98] developed thermoplastic starch (TPS) and linear low-density polyethylene (LLDPE)-based film embedded with 5% sodium nitrite (NaNO_2_) to stabilize oxidation, and microbial growth of vacuum-packaged pork meat stored at 4 °C for 6 days. The result demonstrated that TBARs value, metmyoglobin content, TVC, LABC, and PPC decreased in samples wrapped in active film during the storage. Additionally, the levels of nitrite in meat pork wrapped in NaNO_2_/TPS/LLDPE-based film at approximately 1.05 ppm were lower than the FDA limit (200 ppm) and Chinese (150 ppm) regulation for meat products.

Zhao et al. [100] compared the synergistic effect of antimicrobial mixtures. The first antimicrobial mixture was constituted of 2.5% chitosan and 4.8% carvacrol. The second antimicrobial mixture was composed of 2.5% of chitosan and 10% of gallic acid. Each antimicrobial mixture was incorporated in cassava starch film to inhibit *Listeria monocytogenes* growth in cooked ham during storage at 4 °C for 28 days. The starch film with chitosan and carvacrol fully inhibited *Listeria monocytogenes* growth throughout 4 weeks of storage of cooked ham. The starch film with chitosan and gallic acid had the least effect on cooked ham antimicrobial activity. On the other hand, Wong et al. [102] demonstrated that there was a synergistic effect to reduce microbial growth (TVC) in tilapia fillets (*Orechromis niloticus*) wrapped in low-density polyethylene (LDPE) coated with 1% gallic acid and 1% chitosan when compared to a positive control (chitosan-coated LDPE film) during the storage. Also, there was a synergistic antioxidant effect in wrapped samples in gallic acid-chitosan/LDPE film when compared to negative (LDPE film) and positives (chitosan-coated and gallic-coated LDPE films) controls.

Moreover, Xiong et al. [99] developed a pectin edible coating with the incorporation of 2% of oregano EO and 800 mg/L of resveratrol NEs to extend the shelf life of pork meat stored at 4 °C for 20 days. This active coating inhibited lipid oxidation (TBARs) from 5 to 10 days, and inhibited protein oxidation (DTNB) during the storage of pork meat. Furthermore, microbiological quality (TVC) was extended from 10 to more than 20 days of the study.

Yan et al. [103] prepared a novel antioxidant film incorporating α-tocopherol/chitosan NPs in chitosan/montmorillonite coating film (TOC-CSNPs/CS/MMT film) for sliced dry-cured ham at 4 °C for 120 days. The results indicated that compared to chitosan and chitosan/MMT films, the sliced dry-cured ham coated with TOC-CSNPs/CS/MMT film showed much lower POV and TBARs values during 120 days of storage.

Xin et al. [101] incorporated chitosan-curcumin NPs in zein-potato starch films to prolong the shelf life of *Schizothorax prenati* fillets during storage at −3 °C for 16 days. The result demonstrated that active film functionalized with chitosan-curcumin NPs inhibited the lipid oxidation (TBARs) and protein degradation (TVB-N) of fish fillets prolonging their shelf life by up to 15 days. In addition, Abdou et al. [97] demonstrated similar results in chicken samples coated with pectin and curcumin-cinnamon essential oil NEs. The shelf life of samples with active coating showed the lowest values of total volatile nitrogen (TVN) and TBARs when compared to control samples. Moreover, the microbiological growth of TVC, PPC, and YMC were inhibited in chicken samples coated with pectin and curcumin-cinnamon essential oil NEs.

Amjadi et al. [96] developed gelatin/chitosan nanofiber/ZnO NPs film embedded with 0.4% of betanin NLs to extend the shelf life of fresh beef during storage at 4 °C for 16 days. The result demonstrated that the lipid oxidation of samples wrapped in this active film was inhibited during the storage. In addition, microbial growth of *Staphylococcus aureus* and *Escherichia coli* were inhibited during storage for 16 days.

### 2.5. Active Packaging with Enzymes

Enzymes are biological molecules that play a significant role in an organism’s defense mechanisms against microbial infection. Studies indicated that it is possible to take advantage of the defense mechanism of enzymes in food preservation. Table 3 shows active packaging for muscle foods with the incorporation of enzymes for muscle foods.

Ehsani et al. [66] developed a chitosan-based film containing 10% of lactoperoxidase to preserve the shelf life of carp fish burger stored at 4 °C for 20 days. The results demonstrated that TBARs value was lower than the negative and positive control samples during the storage. Furthermore, samples wrapped in chitosan-based films embedded with lactoperoxidase had lower TVC, PTC, *Pseudomonas* spp., and *Shewanella* spp. counts when compared with other treatments. In addition, Shokri and Ehsani [110] developed whey protein coating functionalized with 2.5% of lactoperoxidase or with 2.5% of lactoperoxidase and α-tocopherol (1.5% and 3%) to prolong the shelf life of Pike–Perch fillets (*Sander lucioperca*, Linnaeus 1758) stored at 4 °C for 16 days. There was no synergistic effect between lactoperoxidase and α-tocopherol to extend the shelf life of fish samples—quite the opposite. Except for the TBA values, which were the lowest using 3% tocopherol, fish samples with whey protein coating with incorporation of 2.5% of lactoperoxidase had lower TVB-N values, and TVC, PTC, *Pseudomonas fluorescens,* and *Shewanella putrefaciens* counts when compared to other treatments.

Wang et al. [111] developed collagen coating embedded with 0.5% of lysozyme to prolong the shelf life of salmon fillets (*Salmo salar*) stored at 4 °C for 15 days. The result showed that treatment with 0.5% lysozyme inhibited the TVB-N values and the TVC in samples with active coating during the storage. On the other hand, Pattarasiriroj et al. [109] wrapped pork belly with rice flour/gelatine/nanoclay film functionalized with 0.5% of catechin-lysozyme and stored at 4 °C for 7 days. The result indicated that wrapping the samples with this active film inhibited the lipid oxidation and decreased microbial growth of TVC and YMC during the storage when compared to the control sample wrapped in PVC.

### 2.6. Active Packaging with Bioactive Peptides and Their Nanocarriers

Bioactive peptides are defined as protein fragments that, when interacting with appropriate receptors, act as antimicrobials [118]. Bioactive peptides can be isolated from food proteins and protein hydrolysates to preserve the shelf life of foods [119]. Antimicrobial peptides show a cationic and amphiphilic behavior due to the attached hydroxyl and amine groups, and hydrophobic alkyl chains [120]. Cationic peptides act on the bacterial surface causing a disturbance in the cell membrane and, in some cases, the peptide enters the target cell [121]. Table 3 shows active meat packaging with the incorporation of bioactive peptides for muscle foods.

Mirzapour-Kouhdasht and Moosavi-Nasab [114] conducted a study evaluating the effects of addition of fish gelatin hydrolysates extracted from *Scomberomorus commerson* skin in active coating (fish skin gelatin or commercial bovine gelatin) on chemical and microbial spoilage in whole shrimp (*Penaeus merguiensis*) stored at 4 °C for 12 days. For both samples, treatments with active coating inhibited TBARs and TVB-B values, and TVC, PTC, LAB growth. Additionally, carbonyl content was inhibited during the storage. In contrast, Rocha et al. [115] found that agar coating functionalized with 0.5% of clove EO to extend the shelf life of flounder (*Paralichthys orbignyanus*) fillets had a greater effect on lipid oxidation and microbiological spoilage than agar coating functionalized with 0.5% of fish protein hydrolysate extracted during the storage at 5 °C for 15 days. On the other hand, the inhibition of *Enterobacteriaceae* and *Pseudomonas* spp. count was higher in samples with fish protein hydrolysate coating.

Maillard reaction products (MRPs) are formed when carbonyl compounds (e.g., reducing sugars) react with free or protein-bound amino acids. MRPs can generate hydrogen peroxide (H_2_O_2_) and reactive oxygen species (ROS) with antimicrobial, and antioxidant activity [122]. In the research by Jiang et al. [113], the incorporation of 0.5 mg/mL squid Maillard peptides in gelatin-chitosan film extended the shelf life of bluefin tuna (*Thunnus thynnus*) stored at 4 °C for 10 days. The squid Maillard peptides were generated from squid waste peptides and D-arabinose. The result showed that POV, TBARs, and TVB-N values for samples wrapped in active film with squid Maillard peptides were significantly lower than those determined for the others. Moreover, TVC was inhibited in sliced bluefin tuna during storage.

Natural ε-polylysine (ε-PL) is an antimicrobial peptide (consists of 25–30 L-lysine residues) with broad antimicrobial activity against Gram-positive and Gram-negative bacteria, molds, and yeasts. ε-PL is water-soluble, biodegradable, stable with low toxicity [123]. This peptide was initially isolated from *Streptomyces albulus* sp. lysinopolymerus strain 346, and later was found in other bacterial or eukaryotic cells. An example is the study by Huang et al. [117], which developed a gelatin–chitosan coating, functionalized with rosemary extract and ε-PL NEs. The coating was designed to inhibit oxidation, and microbial spoilage of ready-to-eat Carbonado chicken meat during storage at 4 °C for 16 days. The result revealed that NEs-coated Carbonado chicken was prolonged by at least 6 days compared to that of the control. Similar results were obtained by Lin, Zhu, and Cui [116], who embedded thyme essential oil in β-cyclodextrin/ε-PLN in the coating of chicken soup gelatin nanofibers stored at 4 °C for 7 days and at 25 °C for 5 days. The microbial growth of *Campylobacter jejuni* in chicken soup was inhibited, to a greater extent, in samples stored at 4 °C than at 25 °C during the storage. Moreover, Alirezalu et al. [112] assessed the effects of ε-PL coating (0.5 and 1%) and stinging nettle extract (3, 6, and 9%) on the shelf life of beef meat. They observed that beef samples coated with 1% of ε-PL and 9% of stinging nettle extract had significantly lower TBARs and TVB-N values compared to the control. Furthermore, it showed the highest effects against YMC, TVC, and TCB during the storage.

Nisin is a bacteriocin produced by *Lactococcus lactis* subsp. lactis that has been widely exploited as a food preservative. Liu et al. [47] investigated the addition of 0.6% NEs of the mixture of antimicrobial peptides (nisin and polylysine) and star anise essential oil in coating with *Artemisia sphaerocephala* Krasch gum of Yao ready-to-eat pork meat stored for 20 days at 4 °C. The microbial spoilage (TVC and *Escherichia coli*) and protein degradation (TVB-N) was retarded in samples with NEs coating during the storage. In contrast, Xiong et al. [93] found that chitosan/gelatin coating functionalized with 0.5% grape seed extract and 0.1% nisin had the worst TBARs and DTNB values in pork meat during the storage at 4 °C for 20 days, when compared to the coated sample with grape seed extract.

### 2.7. Active Packaging with Surfactants

Lauric arginate (LAE, ethyl-Nα-lauroyl-L-arginate hydrochloride) is a derivative of lauric acid, L-arginine, and ethanol. LAE is a promising food preservative with a broad spectrum of antimicrobial activity against Gram-negative and Gram-positive bacteria, fungi, and yeasts due to its chemical properties as a cationic surfactant. LAE also carries a positive charge that influences both its antimicrobial activity and its ability to bind with anionic food components which both impact its antimicrobial properties when applied in food systems [124]. Table 4 shows surfactant-based films used in muscle foods.

Hassan and Cutter [125] investigated the effect of the addition of 0.5, 1.0, or 2.5% of LAE in pullulan-coated polyethylene to preserve the microbiological quality of chicken raw beef, raw chicken breast, and ready-to-eat turkey breast slices stored at 4 °C for 28 days. Sliced raw beef samples wrapped in pullulan-coated polyethylene functionalized with 2.5% of LAE inhibited the growth of *Escherichia coli* to a greater extent during 28 days of shelf life when compared to control sample and samples coated with lower concentrations of LAE. Similar antimicrobial activity of pullulan composite films with addition of LAE were found in sliced raw chicken, against Salmonella, and in ready-to-eat turkey breast, against *Listeria monocytogenes* and *Staphylococcus aureus* cocktails.

On the other hand, there may be an antimicrobial synergistic effect mixing LAE with other natural antimicrobials. In this regard, Pattanayaiying et al. [126] conducted a study, in which they found that, individually, LAE and nisin Z were less efficient than the combination of both. The incorporation of 2% of LAE and 320 AUmL^−1^ nisin Z in pullulan film increased inhibition of microbiological growth in sliced raw turkey breast, ham, and raw beef samples during storage at 4 °C for 28 days. The *Salmonella* Typhimurium ATCC 14028 and *Salmonella* Typhimurium ATCC 13331 growths in raw turkey breast slices wrapped in active film were inhibited during the refrigerated storage. In ham slice samples, the *Staphylococcus aureus* and *Listeria monocytogenes* Scott A growth were reduced immediately. A similar result occurred in raw beef slice samples, since the *Escherichia coli* O157:H7, O111, and O26 growth were reduced immediately after exposure. Similar results were obtained in another study conducted by Pattanayaiying et al. [127]. The authors developed thermoplastic starch/polybutylene adipate terephthalate film with gelatin coating containing 0.8 mg/cm^2^ of LAE, alone or in combination with 69.4 AU/cm^2^ of nisin Z to extend the shelf life of seafood. Microbial spoilage by *Vibrio parahaemolyticus* and *Salmonella* Typhimurium was measured in bigeye snapper (*Lutjanus lineolatus*) and tiger prawn (*Penaeus monodon*) slices during the cooling (28 days at 4 °C) and freezing (90 days at −20 °C) storage. The results revealed that the active film embedded with LAE and nisin/LAE displayed excellent inhibition against *S.* Typhimurium and *V. parahaemolyticus* on chilled and frozen seafood during storage.

### 2.8. Active Packaging with Bacteriophages

Bacteriophages, or phages, are viruses with an average size of 20–200 nm. Phages can be used as natural antimicrobials to extend the shelf life of perishable foods [131]. Table 4 shows phages-based films used in muscle foods.

Alves et al. [129] conducted a study evaluating the effects of 10^8^ PFU/mL of ϕIBB-PF7A bacteriophage onto sodium alginate-based films crosslinked with calcium chloride to prevent chicken breast fillets spoilage caused by *Pseudomonas fluorescens* during storage at 4 °C for 7 days. Phage-containing films applied on commercial chicken fillets were able to control *Pseudomonas fluorescens* growth for a period of up to 5 days.

Phages lack stability in the presence of acidic compounds, enzymes, and evaporated materials, resulting in the loss of antibacterial activity. Moreover, to overcome these challenges, liposomes were introduced to enhance the stability of phages. In this sense, Cui, Yuan and Lin [128] compared the antimicrobial activity of liposome-encapsulated phage (10^11^ PFU/mL) and not encapsulated against *Escherichia coli* O157:H7 in chitosan films to extend the shelf life of beef stored at 25 °C for 7 days. The result indicated that chitosan film containing liposome-encapsulated phage showed positive antibacterial activity against *Escherichia coli* O157:H7 in beef pieces. Additionally, the encapsulation of liposomes increased the stability of phages compared to the negative control sample and non-encapsulated phage sample.

Phages have a proven track record as antimicrobials but should be successfully integrated with active packaging to avoid serious loss of activity. For this, the active packaging should provide the best conditions for a controlled release of antibacterial activity. In this sense, Radford et al. [130] investigated the effects of two bacteriophages (10^12^ PFU/mL of *Salmonella* phage Felix O1 and *Listeria* phage A511 lysate) based xanthan coatings on poly(lactic acid) film to reduce survival and growth of diverse cocktails of *Salmonella* sp. and *Listeria monocytogenes* in precooked sliced turkey breast over 30 days of anaerobic packaging at 4 and 10 °C, and for 14 days in aerobic conditions at 4 and 10 °C. The result indicated that bacteriophage embedded in xanthan-based active packaging coatings can significantly reduce the growth of *Listeria monocytogenes* on aerobically and anaerobically packaged food, whereas the survival of *Salmonella* sp. can be significantly decreased under temperature abuse or prolonged storage by xanthan embedded phage active packaging.

## 3. Future Trends in the Development of Active Packaging for Muscle Foods

From a forward-looking perspective, active packaging for muscle foods will play a vital role in the meat industry as a safety-friendly packaging. The incorporation of antioxidant and antimicrobial compounds in active packaging significantly improves oxidative stability and microbiological safety, extending the shelf life of muscle foods and reducing the amount of antioxidants and antimicrobials used in meat processing. Nonetheless, these active compounds may cause drastic changes in the properties of the polymeric matrix, including their mechanical strength, permeability, volatility, optical and thermal properties, and even their physical appearance. Therefore, it is important to carry out detailed migration studies of the novel active packaging developed for muscle foods under different storage conditions (e.g., humidity, temperature, acidity, among others), and at different storage times to calculate the risk of migration of undesirable compounds to muscle foods. To overcome this challenge, it is important to immobilize these active compounds onto polymers with the help of covalent or ionic bonds. Another approach to reduce the risk associated with the use of synthetic or metal-based nanoparticles is to continue investigating the antimicrobial and antioxidant properties of nanoparticulated materials derived from foods that are safe for the consumer (essential oils, acids, polysaccharides, proteins, etc.) [132,133]. It may be the safest and most acceptable route for the development of novel active packaging for muscle foods. Additionally, active packaging does not provide visual information indicative of the safety and quality of foods during the shelf life. In this regard, another type of packaging—intelligent packaging—was recently developed, which can convey detailed information about the condition of a packaged food or its environment throughout a logistical chain, as well as provide early warning to the food manufacturer and to the consumer. Therefore, nanotechnology, together with active, intelligent, and eco-friendly packaging technologies, can work synergistically to produce the best packaging system without negative interactions between components [134]. It is expected that, soon, the application of these novel technologies in packaging will improve the preservation, distribution, availability, and consumption of muscle foods from the perspective of population growth.

## 4. Conclusions

Active packaging for muscle foods is constantly improving. Due to concerns about reducing the direct application of additives to muscle foods to extend shelf life, an expressive effort has been made by the food industry. Research results have shown how non-conventional active packaging can improve the oxidation and microbial stability in muscle foods. The use of nanotechnology to encapsulate consumer-safe active substances with antioxidant and antimicrobial properties has demonstrated high potential in the development of novel active packaging for muscle foods. On the other hand, it is necessary to investigate the behavior of active substances applied to muscle food packaging under different conditions of temperature, acidity, pH, humidity, storage time, among others, to better understand their kinetics and potential danger. This review suggests that the sustainable development of novel active packaging technologies using nanotechnology can provide new opportunities in muscle food preservation techniques.

## Figures and Tables

**Figure 1 foods-12-03662-f001:**
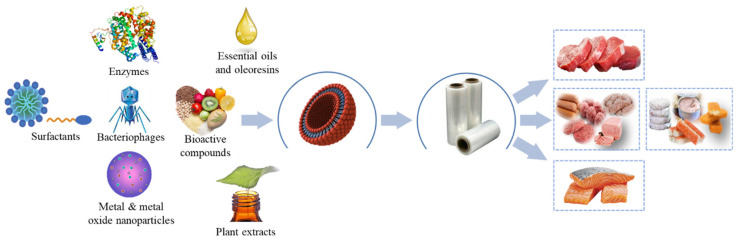
Application of active packaging using nanotechnology in muscle foods.

**Figure 2 foods-12-03662-f002:**
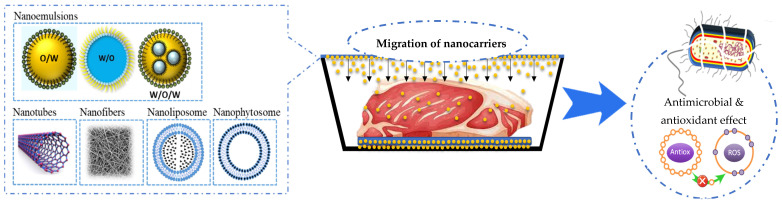
Types of non-conventional active packaging with nanocarriers for muscle foods.

**Figure 3 foods-12-03662-f003:**
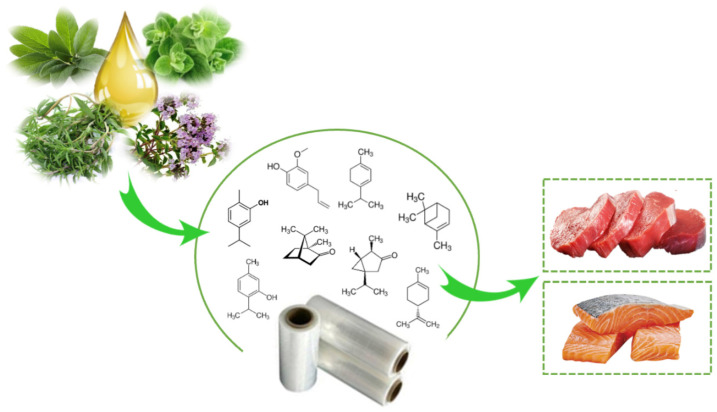
Active packaging with incorporation of essential oil for muscle foods.

**Figure 4 foods-12-03662-f004:**
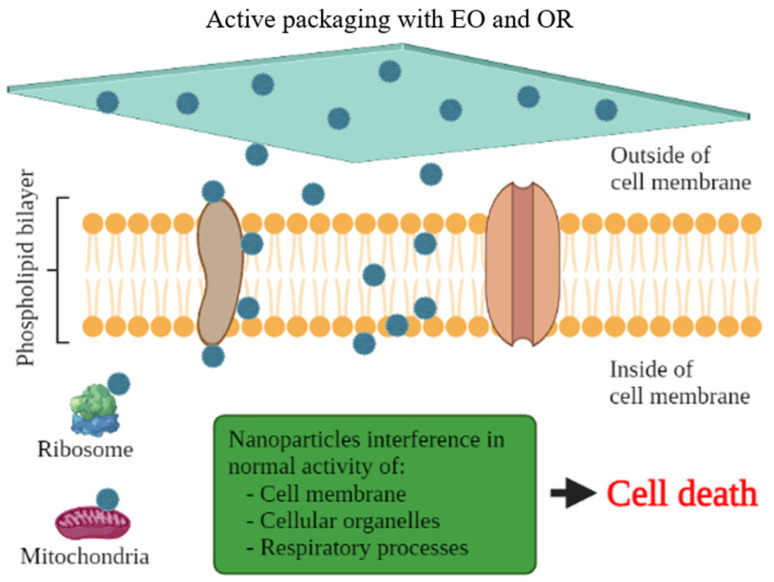
Cell death of microorganisms through the action of antimicrobial substances in active packaging with essential oils (EO) and oleoresins (OR) for muscle foods.

**Figure 5 foods-12-03662-f005:**
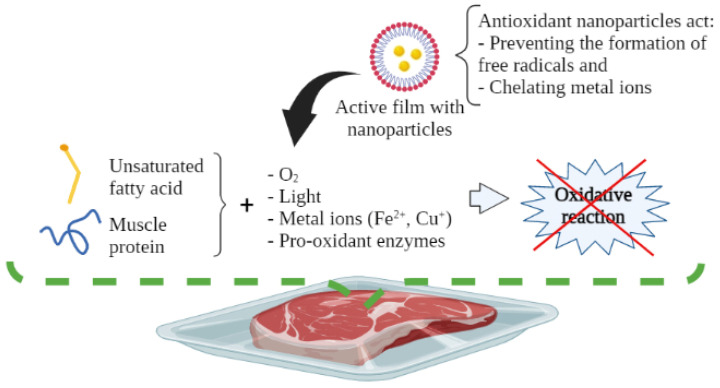
Antioxidant mechanism of active packaging with nanoparticles for muscle foods.

**Table 2 foods-12-03662-t002:** Active packaging functionalized with extract and other active compounds, and their nanocarriers for muscle foods.

Meat/Meat Products	Extract and Other Active Compounds	Polymeric Matrix	Inhibitory Effect Against	Reference
Fish fillets	Amaranthus leaf extract	PVA/gelatin film	Lipid oxidation, protein degradation, TBC, *Staphylococcus aureus* and fecal coliforms	[88]
Minced chicken meat	Amaranthus leaf extract	PVA/gelatin film	Lipid oxidation, protein degradation, TBC, *Staphylococcus aureus* and fecal coliforms	[88]
Chicken meat	Lemon peel polyphenol extract	Chitosan or pullulan coating	Lipid oxidation, TBC, and ENT	[90]
Chicken fillets	Grape seed extract	Chitosan/nanocellulose film	Lipid oxidation, TVC, and TCB	[91]
Pork meat	Pomegranate peel extract nanoencapsulation	Zein film	*Listeria monocytogenes*	[92]
Pork meat	Grape seed extract	Chitosan–gelatine edible coating	Lipid and protein oxidation	[93]
Minced beef	Green tea extract	Polyamide film	Lipid and protein oxidation	[94]
Sliced salami	Rambutan peel extract-cinnamon oil	Whey protein isolate film	TVC	[95]
Beef	Betanin nanoliposomes/ZnO NPs	Gelatin/chitosan nanofibers film	Lipid oxidation, *Staphylococcus aureus*, and *Escherichia coli*	[96]
Chicken meat	curcumin-cinnamon essential oil NEs	Pectin coating	Lipid oxidation, protein degradation, and TVC, PPC, YMC	[97]
Pork meat	Sodium nitrate	LLDPE/TPS film	Lipid and protein oxidation, TVC, LABC, and PPC	[98]
Pork meat	Resveratrol/oregano essential oil NEs	Pectin coating	Lipid and protein oxidation, TVC	[99]
Cooked ham	Carvacrol/chitosan	Cassava starch film	*Listeria monocytogenes*	[100]
Fish fillets	Chitoson/curcumin NPs	Zein/potato starch film	Lipid oxidation and protein degradation	[101]
Fish fillets	Chitosan/gallic acid	Coated LDPE film	Lipid and protein degradation, and TVC	[102]
Sliced dry-cured ham	α-tocopherol/chitosan NPs	Chitosan/montmorillonite coating	Lipid oxidation	[103]

ENT, *Enterobacteriaceae*, LABC, lactic acid bacteria count; LDPE, low-density polyethylene; LLDPE; linear low-density polyethylene; NEs, nanoemulsions; NPs, nanoparticles; PPC, psychrophilic count; PVA; polyvinyl alcohol; TBC, total bacterial count; TCB, total coliform bacteria; TPS, thermoplastic starch; TVC, total viable count; YMC, yeast and mold count.

**Table 3 foods-12-03662-t003:** Studies considering antioxidant and antimicrobial effect of active meat packaging with addition of enzymes and peptides, and their nanocarriers in muscle foods.

Meat/Meat Products	Enzymes and Other Peptides	Polymeric Matrix	Inhibitory Effect Against	Reference
Ready-to-eat Yao pork meat	Nisin, polylysine and star anise essential oil NEs	*Artemisia sphaerocephala* Krasch. gum coating	Protein oxidation and protein degradation	[47]
Fish burger	Lactoperoxidase	Chitosan film	Lipid oxidation, TVC, PTC, *Pseudomonas* spp., and *Shewanella* spp.	[66]
Pork meat	Nisin	Chitosan/gelatin coating	Lipid and protein oxidation, and TVC	[93]
Pork belly	Catechin-lysozyme	Rice flour–gelatine–nanoclay film	Lipid oxidation and TVC	[109]
Fish fillets	Lactoperoxidase	Whey protein coating	Protein degradation, TVC, PTC, *Shewanella putrefaciens*, and *Pseudomonas fluorescens*	[110]
Fish fillets	Lactoperoxidase/α-tocopherol	Whey protein coating	Lipid oxidation	[110]
Fish fillets	Lysozyme	Collagen coating	Protein degradation and TVC	[111]
Beef	ε-polylysine/mixed plant extracts	ε-polylysine coating	YMC, TVC, and TCB	[112]
Fish meat	Maillard peptides	Gelatin/chitosan film	Lipid oxidation, protein degradation, and TVC	[113]
Whole shrimp	Fish skin gelatin hydrolysates	Fish skin gelatin or commercial bovine gelatin coating	Lipid and protein oxidation, protein degradation, TVC, PTC, and LABC	[114]
Fish fillets	Fish protein hydrolysate or clove essential oil	Agar film	TVB-N, TVC, H_2_S-producing organisms, *Pseudomonas* spp., ENT, and LABC	[115]
Chicken soup	Thyme essential oil/β-cyclodextrin/ε-polylysine NPs	Gelatin nanofiber coating	*Campylobacter jejuni*	[116]
Ready-to-eat Carbonado chicken meat	Rosemary extract/ε-polylysine NEs	Gelatin/chitosan coating	Lipid oxidation, protein degradation, TVC, YMC and ENT	[117]

ENT, *Enterobacteriaceae;* LABC, lactic acid bacteria count; NEs, nanoemulsions; NPs, nanoparticles; PTC, psychrotrophic count; TVC, total viable count; YMC, yeast and mold count.

**Table 4 foods-12-03662-t004:** Active packaging functionalized with surfactants and phages used for muscle foods.

Meat/Meat Products	Surfactants and Phages	Polymeric Matrix	Inhibitory Effect Against	Reference
Sliced beef	Lauric arginate	Pullulan-coated PE	*Escherichia coli* cocktail	[125]
Sliced chicken meat	Lauric arginate	Pullulan-coated PE	*Salmonella* cocktail	[125]
Sliced turkey meat	Lauric arginate	Pullulan-coated PE	*Listeria monocytogenes* and *Staphylococcus aureus* cocktail	[125]
Sliced turkey meat	Lauric arginate/nisin Z	Pullulan film	*Salmonella* Typhimurium and *Salmonella* Enteritidis	[126]
Sliced beef	Lauric arginate/nisin Z	Pullulan film	*Escherichia coli* O157:H7, O111 and O26	[126]
Sliced ham	Lauric arginate/nisin Z	Pullulan film	*Staphylococcus aureus* and *Listeria monocytogenes*	[126]
Sliced fish	Lauric arginate/nisin Z	Coated TPS/PBAT film	*Vibrio parahaemolyticus* ATCC 17802 and *Salmonella* Typhimurium ATCC 1402	[127]
Beef	Liposome-encapsulated phage	Chitosan film	*Escherichia coli* O157:H7	[128]
Chicken fillet	Phages φIBB-PF7A	Sodium alginate-based film	*Pseudomonas fluorescens*	[129]
Ready-to-eat sliced turkey	Listeria phage A511	Xanthan gum-coated PLA	*Listeria monocytogenes*	[130]

PBAT, polybutylene adipate terephthalate; PE, polyethylene; PLA, poly(lactic acid); TPS, thermoplastic starch.

## Data Availability

The data used to support the findings of this study can be made available by the corresponding author upon request.

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
