# Peer review of "Recent Trends in Active Packaging Using Nanotechnology to Inhibit Oxidation and Microbiological Growth in Muscle Foods"

_foods, 2023, doi:10.3390/foods12193662_

Round 1

Reviewer 1 Report

Comments and Suggestions for Authors

Reviewer #: This review introduced the role of nanotechnology in preserving the muscle food. Particularly, the applications of various nanomaterials in the active packaging for the inhibition of oxidation and microbial growth in the muscle food.  

1) A review article aims at achieving THREE essential objectives:

i) providing a complete, structured and systematic summarization on the related key aspects. This means that the authors will summarize those in many figures and tables;

Objectives 1) has been successfully achieved. The review contains sufficient number of figures and tables which highlighted the growth of the field.

ii) presenting new discoveries from the authors’ own knowledge synthesis based on existing literature results. This means that the authors will provide important and synthesized new knowledge that are not included in those articles in the literature;

Objective ii) This objective has been successfully achieved by incorporating latest and related literature.

iii) outlining detailed views on future research directions and perspectives.

Objective iii) This has been adequately addressed in the section 3 about the future trends in the development of active packaging.

However, there are certain minor points which authors need to address, such as,

In the abstract, types of nanomaterials used for the purpose should be included and few more benefits of nanomaterials in the active packaging of muscle food should be discussed.

2) At the end of “Introduction” section, the authors need to justify why the review article fills a critical gap in the field, is indeed in need and timely.

Figures 4 and 5 is about the antibacterial and antioxidant mechanism of active packaging with nanoparticles for muscle foods, but they are given and discussed in the section which deals with essential oils and oleoresins, this requires an explanation. OR these figures should be cited at the proper places.

Comments on the Quality of English Language

minor changes are required

Author Response

Reviewer 1

Reviewer #: This review introduced the role of nanotechnology in preserving the muscle food. Particularly, the applications of various nanomaterials in the active packaging for the inhibition of oxidation and microbial growth in the muscle food.

A review article aims at achieving THREE essential objectives:

  1. i) providing a complete, structured, and systematic summarization on the related key aspects. This means that the authors will summarize those in many figures and tables;

Objectives i) has been successfully achieved. The review contains sufficient number of figures and tables which highlighted the growth of the field.

  1. ii) presenting new discoveries from the authors’ own knowledge synthesis based on existing literature results. This means that the authors will provide important and synthesized new knowledge that are not included in those articles in the literature;

Objective ii) This objective has been successfully achieved by incorporating latest and related literature.

iii) outlining detailed views on future research directions and perspectives.

Objective iii) This has been adequately addressed in the section 3 about the future trends in the development of active packaging.

However, there are certain minor points which authors need to address, such as,

1) In the abstract, types of nanomaterials used for the purpose should be included and few more benefits of nanomaterials in the active packaging of muscle food should be discussed.

Response: Thank you for your consideration. Changes were included in the abstract (lines 25-31).

2) At the end of “Introduction” section, the authors need to justify why the review article fills a critical gap in the field, is indeed in need and timely.

Response: The end of the “Introduction” was rewritten as suggested by reviewer (lines 148-158).

3) Figures 4 and 5 is about the antibacterial and antioxidant mechanism of active packaging with nanoparticles for muscle foods, but they are given and discussed in the section which deals with essential oils and oleoresins, this requires an explanation. OR these figures should be cited at the proper places.

Response: Changes were made in Figure 4 (line 194). And, Figure 5 was relocated in the manuscript (line 365).

4) Comments on the Quality of English Language

minor changes are required

Response: Thank you for your consideration. The manuscript was revised by a native English speaker.

Reviewer 2 Report

Comments and Suggestions for Authors

Jacinto-Valderrama et al. provided an overview of active packaging for muscle foods, focusing particularly on the antimicrobial and antioxidant attributes. The study offers a comprehensive examination along with mechanistic pathways of active agents through which active agents function to preserve the quality of packaged items. However, the paper's organization and discussion require enhancement to ensure a more coherent and effective presentation. Several comments are outlined that necessitate improvement before further consideration by the editor

General comments

- Paper should be linguistically improved.

- The schematic descriptions along with photo images of reviewed studies should be provided for a better understanding of the performance of the types of packaging discussed in the paper.

- How new technologies can shape the future of meat active packaging e.g. Advanced Functional Materials. 2021 Jul;31(28):2010759.

Abstract

- ‘Active packaging” should be added to keywords.

Introduction & context

- Lines 98-99. The active agents do not necessarily release from packaging in all scenarios. There are certainly some types of active packaging posing controlled release behavior, but it is not always the case. So, please moderate the sentence.

- Lines 100-102- Please revise this sentence. The intended meaning is not clear due to grammatical errors.

- Lines 103-105. Can you specify what types of antioxidants have been added to polymers using extrusion methods? There might be some doubt regarding low thermal stable active agents e.g. essential oils.

- Lines 126-127. …. With nanoscale dimensions in at least one of the dimensions.

- Lines 129-131. The sentence is not grammatically correct.

- The authors should provide a concise overview of prior reviews on active packaging for meat and distinctly highlight the unique contributions and originality of their own review.

- The paper should clearly address its objectives, detailing how it aims to address existing gaps in the literature. Furthermore, it should clarify how the identified gaps are being addressed and filled through the content of the paper.

- Figure 4 schematically shows the mechanistic pathway of active packaging derived from nanoparticles while the context discusses the EO and RO. Please use the relevant context and figures to keep the coherence of the paper.

- How does the addition of active agents to meat packaging impact the marketability, consumer health, and primary properties of packaging? What are the future directions and new technologies in this field? Should be briefly discussed. There are many new technologies to overcome the migration of active agents e.g. Reactive and Functional Polymers. 2021 Jan 1;158:104792; Surfaces and Interfaces. 2021 Feb 1;22:100814; Progress in Organic Coatings. 2022 Jan 1;162:106556. Additionally, there are some other technologies that can be used as active packaging, especially in direct contact e.g. Surfaces and Interfaces. 2022 Feb 1;28:101573.

- The concluding section needs enhancement to accurately capture the paper's context and findings.

Comments on the Quality of English Language

The language is marginally acceptable but it needs to be improved for better readability and flowbility. 

Author Response

Reviewer 2: 

Jacinto-Valderrama et al. provided an overview of active packaging for muscle foods, focusing particularly on the antimicrobial and antioxidant attributes. The study offers a comprehensive examination along with mechanistic pathways of active agents through which active agents function to preserve the quality of packaged items. However, the paper's organization and discussion require enhancement to ensure a more coherent and effective presentation. Several comments are outlined that necessitate improvement before further consideration by the editor.

General comments

1) Paper should be linguistically improved.

Response: Thank you for your consideration. The manuscript was revised by a native English speaker.

2) The schematic descriptions along with photo images of reviewed studies should be provided for a better understanding of the performance of the types of packaging discussed in the paper.

Response: The descriptions of Figures and their titles have been changed for better understanding as suggested by the reviewer.

3) How new technologies can shape the future of meat active packaging e.g. Advanced Functional Materials. 2021 Jul;31(28):2010759.

Response: This point was addressed in the topic “Future Perspectives” (lines 662-670).

4) Abstract: ‘Active packaging” should be added to keywords.

Response: “Active packaging” was added to Keywords (line 35).

5) Introduction & context

  1. i) Lines 98-99. The active agents do not necessarily release from packaging in all scenarios. There are certainly some types of active packaging posing controlled release behavior, but it is not always the case. So, please moderate the sentence.

Response: The Reviewer’s suggestion was accepted and the sentence was changed (lines 99).

  1. ii) Lines 100-102- Please revise this sentence. The intended meaning is not clear due to grammatical errors.

Response: It was clarified this sentence in the manuscript (lines 101-102).

iii) Lines 103-105. Can you specify what types of antioxidants have been added to polymers using extrusion methods? There might be some doubt regarding low thermal stable active agents e.g. essential oils.

Response: Changes were made on lines 205-209 to specify the types of antioxidants incorporated into coatings and films produced by extrusion. A new sentence was added to briefly address low thermal stable active agents.

  1. iv) Lines 126-127. …. With nanoscale dimensions in at least one of the dimensions.

Response: The authors included the reviewer´s suggestion in the manuscript (lines 129).

  1. v) Lines 129-131. The sentence is not grammatically correct.

Response: Changes were made in the sentence as suggested by the reviewer (lines 130-133).

  1. vi) The authors should provide a concise overview of prior reviews on active packaging for meat and distinctly highlight the unique contributions and originality of their own review.

Response: This point was addressed at the end of the “Introduction” (lines 150-158).

vii) The paper should clearly address its objectives, detailing how it aims to address existing gaps in the literature. Furthermore, it should clarify how the identified gaps are being addressed and filled through the content of the paper.

Response: Thank you for your consideration. This point was rewritten at the end of the “Introduction” (lines 154-158).

viii) Figure 4 schematically shows the mechanistic pathway of active packaging derived from nanoparticles while the context discusses the EO and RO. Please use the relevant context and figures to keep the coherence of the paper.

Response: Changes were made in Figure 4 in order to improve understanding (lines 193).

  1. ix) How does the addition of active agents to meat packaging impact the marketability, consumer health, and primary properties of packaging? What are the future directions and new technologies in this field? Should be briefly discussed. There are many new technologies to overcome the migration of active agents e.g. Reactive and Functional Polymers. 2021 Jan 1;158:104792; Surfaces and Interfaces. 2021 Feb 1;22:100814; Progress in Organic Coatings. 2022 Jan 1;162:106556. Additionally, there are some other technologies that can be used as active packaging, especially in direct contact e.g. Surfaces and Interfaces. 2022 Feb 1;28:101573.

 Response: Thank you for your consideration. The objective of this work was to present the results of the antioxidant and antimicrobial effect of the application of non-conventional technologies, including nanotechnology, in active packaging for muscle foods. Furthermore, it is hoped that this review will serve as a basis for other researchers who are delving into the world of active packaging in muscle foods. Novel technologies in active packaging that have not yet been applied to muscle foods were not addressed in this work.

  1. x) The concluding section needs enhancement to accurately capture the paper's context and findings.

Response: Changes were made in the “Abstract”, at the end of “Introduction” and the “Future trends” as suggested by the reviewer.

6) Comments on the Quality of English Language

The language is marginally acceptable, but it needs to be improved for better readability and flowbility. 

Response: The manuscript was revised by a native English speaker.

Reviewer 3 Report

Comments and Suggestions for Authors

In this review article, the authors summerized commonly used active substances in active packaging polymer matrix, and discussed the leading role of nanotechnology in enhancing the antioxidant and antibacterial effects of these active substances. In addition, the extension of shelf-life in muscle food and the inhibition effect of oxidation and microbial growth during storage were reviewed. 

Here are several questions and/or suggestions the authors may consider:

1. In the introduction, the background and recent advances of muscle food should be further introduced.

2. The computation of potentially toxicity of natural and synthetic additives should be discussed in detail. I will doubt this idea that all natural additives are safer than synthetic additives. 

3. The conclusion of the article needs to be improved. The summary should not only describe and analyze the previous studies, but also include the author's opinions and perspective.

4. There are some format errors in the reference list. eg. ref 6, ref 11, ref 44, ref 51, ref 54, ref 65, etc

Author Response

In this review article, the authors summerized commonly used active substances in active packaging polymer matrix and discussed the leading role of nanotechnology in enhancing the antioxidant and antibacterial effects of these active substances. In addition, the extension of shelf-life in muscle food and the inhibition effect of oxidation and microbial growth during storage were reviewed. 

Here are several questions and/or suggestions the authors may consider:

  1. In the introduction, the background and recent advances of muscle food should be further introduced.

Response: Thank you for your consideration. In the Introduction, in lines 101-103, 113-119 and 128-138, conventional applications found today in retail were discussed. Presenting new alternatives is still little explored by the muscle food industry. However, concluding that future novel technologies in active packaging for muscle foods must consider the possible dangers in the sensorial modification of the food, and the environment.

  1. The computation of potentially toxicity of natural and synthetic additives should be discussed in detail. I will doubt this idea that all natural additives are safer than synthetic additives. 

Response: Lines 115-116 describe that natural additives that do not bring side effects to consumers are an alternative for application in active packaging with antioxidant and antimicrobial effects. Furthermore, lines 172, 308, 319-320, 342, 378, 549 address the degree of toxicity of some of the natural and synthetic active substances addressed in this work.

  1. The conclusion of the article needs to be improved. The summary should not only describe and analyze the previous studies, but also include the author's opinions and perspective.

Response: Changes were made to the “Conclusion” as suggested by the reviewer (lines 677-684).

  1. There are some format errors in the reference list. eg. ref 6, ref 11, ref 44, ref 51, ref 54, ref 65, etc

Response: Changes were made as described by the Foods journal.
